# Physical, Psychological, and Body Composition Differences between Active and Sedentary Adolescents According to the “Fat but Fit” Paradigm

**DOI:** 10.3390/ijerph191710797

**Published:** 2022-08-30

**Authors:** Adrián Mateo-Orcajada, Raquel Vaquero-Cristóbal, Francisco Esparza-Ros, Lucía Abenza-Cano

**Affiliations:** 1Faculty of Sport, Catholic University San Antonio of Murcia, 30107 Murcia, Spain; 2Kinanthropometry International Chair, Catholic University San Antonio of Murcia, 30107 Murcia, Spain

**Keywords:** adolescent health, anthropometric measurement, nutritional habits, physical activity, weight status

## Abstract

The practice of physical activity during adolescence is essential for the proper development of the population. In recent decades, the relevance of physical activity has been increasing, due to the development of the “fat but fit” paradigm. This paradigm shows that adolescents with a high level of physical fitness are healthier than adolescents with poorer physical fitness, regardless of their weight, giving importance to sports practice over other aspects. However, few previous studies have analyzed the differences in physical and body composition between active and sedentary adolescents in this paradigm. For this reason, the objectives of the present study were to establish the differences in body composition, physical performance, and adherence to the Mediterranean diet between active and sedentary adolescents; and to analyze the differences between active and sedentary adolescents according to the “fat but fit” paradigm. The sample consisted of 791 adolescent whose body composition, level of physical activity, adherence to the Mediterranean diet, and physical fitness were measured. It was found significant between active and sedentary adolescents in most of the anthropometric, AMD, and physical fitness variables, with a significant effect of the covariates gender, age, BMI, and biological maturation on the model. The binary logistic regression analysis performed shows that anthropometric variables, AMD, and VO2 max can be considered as primary outcomes to distinguish between active and sedentary groups of adolescents. Furthermore, the results showed that the active adolescents, regardless of their weight status, had lower fat mass and greater muscle mass, as well as a higher performance in the physical fitness tests, and greater adherence to the Mediterranean diet than the sedentary adolescents. To conclude, the practice of physical activity is a determinant for the improvement of body composition, physical performance, and adherence to the Mediterranean diet of the adolescent population, regardless of their gender, age, weight, or maturity status.

## 1. Introduction

The practice of physical activity during adolescence is fundamental for the prevention and treatment of different chronic diseases [1], among which obesity [2], hypertension [3], diabetes [4], or metabolic syndrome [5] stand out. Despite its relevance, the level of physical practice is annually reduced by 3.4% in boys and 5.3% in girls from the age of nine [6]. The decrease in the level of daily physical activity and the adoption of sedentary behaviors are related to lower adherence to the Mediterranean diet (AMD) [7], lower performance on physical fitness tests [8], and higher percentage of fat mass [9]. This forms a serious problem for the health of the adolescent population because in recent years there has been a decline in physical capacities, mainly affecting strength [10] and cardiorespiratory capacity [11,12], as well as an increased number of diseases related to poor nutrition [13] and accumulation of fat mass [14].

Unfortunately, the reasons why the adolescent population has decreased the practice of physical activity and adherence to certain healthy habits are very diverse. Among them, the barriers found for the practice of physical activity, among which fatigue, obligations, lack of time, and environment/facilities stand out [15]; the COVID-19 pandemic experienced in recent years has not helped either, decreasing the level of physical activity and sports practice of the adolescent population during and after lockdown [16,17]; and the emergence of new forms of leisure that favor an increase in sedentary time and decrease the physical activity level and the outdoor time [18]. All these reasons hinder the practice of physical activity, with considerable repercussions on the health of adolescents during this and later stages.

Therefore, the practice of physical activity during adolescence is necessary due to the benefits in physical condition and body composition, with previous research finding improvements in muscle mass and decreases in fat percentage with the practice of daily physical activity, as well as improvements in physical fitness [19]. Physical fitness acquires special relevance in this regard because scores obtained in physical fitness tests during adolescence can predict health-related fitness in adulthood [20], with cardiorespiratory capacity, upper limb strength, speed, and flexibility being the physical capacities most valued in previous research conducted with adolescents due to their relationship with health [21]. So much so that research conducted in recent years has given rise to a phenomenon known as “fat but fit”, in which overweight or obese individuals, but with a high level of cardiorespiratory fitness or a good fitness level, have a lower risk of metabolic and cardiovascular diseases, regardless of their weight status [22,23], giving even more relevance to the practice of physical activity.

Although scientific research conducted to date has demonstrated the importance of the practice of physical activity for adolescents, few studies have analyzed the differences between active and sedentary adolescents, and also have limitations that make it difficult to extrapolate the results to adolescents aged 12–16 years old, such as the use of different methodologies to measure the study variables; the small sample size; the different age ranges; or the inclusion of a disparate number of adolescent boys and girls, or adolescents of only one sex [24].

Regarding the “fat but fit” paradox, few studies have been carried out in adolescents, and these are observational studies that have used cardiorespiratory fitness or handgrip strength as variables to determine physical fitness [22,23]. However, part of the variability in fitness is explained by genetics and hormonal changes, so that adolescents who are not very active can present moderate levels of these variables, which is a factor to be taken into account when considering subjects as fit [25]. One variable that could solve this problem would be the level of physical activity performed, but no previous research is known that has grouped adolescents according to this variable. The level of physical activity is related to adolescents’ health and physical fitness [26] and presents numerous valid and reliable ways of being measured, including electronic devices based on accelerometry, or self-reported questionnaires [27]. With respect to the self-reported questionnaires, the physical activity questionnaire for adolescents (PAQ-A) is the most valid and reliable in the adolescent population [28] and makes it possible to distinguish between active and sedentary adolescents [29].

Therefore, there is a gap in the scientific literature because no previous research has considered the level of physical activity as a variable to determine which adolescents are fit and how this relates to the rest of the health-related variables. Therefore, further scientific research is needed to include the level of physical activity as a discriminating variable, since it provides a great deal of information in this regard and will make it possible to determine the differences between adolescents who are truly active and those who are sedentary. In addition, the limitations found in previous research conducted with adolescents with respect to methodological rigor, non-validated instruments, and reduced samples will be addressed. For these reasons, the objectives of the present study were (a) to establish the differences in body composition, physical performance, and AMD between active and sedentary adolescents; and (b) to analyze the differences between active and sedentary adolescents according to the “fat but fit” paradigm considering different weight status.

Based on the results of previous research, two hypotheses are put forward for the present investigation: (a) active adolescents will show better body composition, physical fitness, and AMD than sedentary adolescents; and (b) considering the “fat but fit” paradigm, and in line with previous research conducted with cardiorespiratory fitness and handgrip strength as discriminatory variables, adolescents who show a higher level of physical activity will present higher physical fitness and also better body composition, regardless of their weight status.

## 2. Materials and Methods

### 2.1. Design

The present study was cross-sectional, with non-probability convenience sampling. Prior to the start of the study, the institutional ethics committee reviewed and approved the protocol designed in accordance with the World Medical Association code (CE022102). The measurement protocol was registered before the start of the study at ClinicalTrials.gov (code: NCT04860128). The research design and the development of the manuscript also followed the STROBE statement [30]. The sample was chosen non-probabilistically by convenience.

### 2.2. Participants

Sample size was calculated using Rstudio 3.15.0 statistical software (Rstudio Inc., Boston, MA, USA) and using standard deviations (SD) from previous research that examined physical activity level (SD = 0.58) [31] of adolescents aged 12–16 years old. The estimated error (d) for a 99% confidence interval was 0.05 for the level of physical activity. The minimum sample necessary for the development of the research was 750 adolescents.

The final sample consisted of 791 adolescents (404 boys and 387 girls) between the ages of 12 and 16 (mean age: 14.39 ± 1.26 years). Of the total sample, 348 adolescents were active, of whom 226 were boys and 122 girls; while 443 adolescents were sedentary, of whom 186 were boys and 257 girls. The average score (from a minimum score of 1 to a maximum score of 5) of physical activity performed was 2.63 ± 0.67 (boys: 2.81 ± 0.67; girls: 2.46 ± 0.62).

In order to obtain a representative sample of the urban areas of Region of Murcia (Spain) and in accordance with the data collected by the Regional Statistics Centre of Murcia [32], which municipality had the largest school-age population of secondary school students in the north, west, east, and south of the Region was determined. After this, based on the Regional Ministry of Education of the Region of Murcia data [33], the high school with the highest number of students enrolled in compulsory secondary education was selected within each of these municipalities. All of them decided to participate in this study on a voluntary basis. After explaining the objectives and the procedure to be followed to the directors of the schools and those responsible for the physical education area, an informative meeting was held with the students and parents of each school to explain the objectives of the research, the questionnaires, and physical fitness tests to be carried out, as well as the confidential treatment of the data. After this, the adolescents who wished to participate voluntarily provided an informed consent signed by them and their parents. Figure 1 shows the sample selection flow chart.

All adolescents who participated in the research met the following inclusion criteria: (a) age between 12 and 16 years old; (b) attending compulsory secondary education; (c) not presenting any incapacitating disease that prevented participation; and (d) completing all the questionnaires and physical tests in their entirety.

### 2.3. Instrumentation

#### 2.3.1. Questionnaire Measures

For data collection, tests were selected that had been previously validated in the adolescent population and used in previous studies. To obtain the level of physical activity of the adolescents, the PAQ-A was used, which had been previously validated in Spanish. It had an intraclass correlation coefficient of 0.71 for the final score of the questionnaire [28]. This questionnaire is a 7-day recall and self-administered questionnaire composed of nine items that assess the level of physical activity performed in the week prior to the study. The first eight items of the questionnaire have a Likert scale of 1–5 points for completion (1: no physical activity; 5: a lot of physical activity), while the ninth item is answered dichotomously (yes or no). The final physical activity score was calculated as the arithmetic mean of the scores from the first eight items, the minimum score being 1 and the maximum score 5. Subsequently, the subjects were classified according to the score obtained in the questionnaire, those with a score higher than 2.75 being active, and those who obtained a lower score were sedentary, as done in previous research [29].

The “Mediterranean Diet Quality Index for children and adolescents” (KIDMED) questionnaire [34] was used to assess the nutritional habits of adolescents, specifically the AMD. This questionnaire is composed of 16 items that are answered with a dichotomous scale (yes or no), and whose score varies between −1 (negative connotation) and +1 (positive connotation). Twelve questions had positive scores and four negative scores, with the total score ranging from 0 to 12 [35].

#### 2.3.2. Body Composition Measurement

The body composition analysis was performed by three accredited International Society for the Advancement of Kinanthropometry (ISAK) anthropometrists (one level 2, one level 3, and one level 4). Three basic measurements (body mass, height, sitting height), three skinfolds (triceps, thigh, and calf), and five girths (arm relaxed, waist, hip, thigh, and calf) were measured according to the protocol standardized by the International Society for the Advancement of Kinanthropometry (ISAK) [36].

The variables were measured twice, with a third measurement being necessary when the difference between the first two measurements was greater than 5% for the skinfolds and 1% in the rest of the measurements. The mean of the measured values, when two measurements were performed, and the median of the values, when three measurements were performed, were used as the final value [36]. All measurements corresponding to each subject were performed by the same anthropometrist.

To measure girths, an inextensible tape, Lufkin W606PM (Lufkin, Missouri City, TX, USA), with a 0.1 cm accuracy was used; a skinfold caliper (Harpenden, Burgess Hill, UK) with an accuracy of 0. 2 mm was used for measuring skinfolds; for body mass, a TANITA BC 418-MA Segmental (TANITA, Tokyo, Japan) with an accuracy of 100 g; and for height and sitting height an SECA stadiometer 213 (SECA, Hamburg, Germany) with an accuracy of 0.1 cm was used. All the instruments were previously calibrated.

The intra- and inter-evaluator technical error of measurements (TEM) were calculated in a sub-sample. The intra-evaluator TEM was 0.02% for the basic measurements; 1.21% for skinfolds, and 0.04% for the girths; and the inter-evaluator TEM was 0.03% for the basic measurements; 1.98% for skinfolds and 0.06% for the girths.

The final values of the anthropometric measurements were used to calculate the BMI, fat mass (%) [37], muscle mass [38], Σ3 skinfolds (triceps, thigh, and calf), waist-to-hip ratio (waist girth/hip girth) [39], and waist-to-height ratio (waist girth/height) [40]. Muscle girths were estimated by correcting limb girths for the appropriate skinfold using a circular model of the limb cross-section and assuming that the adipose tissue thickness was half the skinfold thickness [41,42]. Thus, the corrected girths of the arm [arm relaxed girth—(π * triceps skinfold)], thigh [middle thigh girth—(π * thigh skinfold)], and calf [calf girth—(π * calf skinfold)] were calculated.

The sex-specific formula from Mirwald et al. [43] was used to estimate the maturity offset of the adolescents. From the maturity offset, the biological maturation of each subject was calculated using the formula: biological maturation = chronological age—maturity offset result [44]. This method proved to be valid for estimating the maturity offset with respect to the gold standard using regression equations with an R^2^ = 0.92–0.89 in the case of boys and an R^2^ = 0.91–0.88 in the case of girls [45].

#### 2.3.3. Physical Fitness Test

Regarding physical condition, the sit-and-reach test was used to measure hamstring flexibility [46] (Figure 2). The participants were seated with knees extended, feet hip-width apart, ankles at 90° flexion, toes pointed upward, and the sole of the foot fully supported on an Acuflex Tester III box (Novel Products, Rockton, IL, USA). From that position, the subjects had to perform a maximum trunk flexion, keeping the knees and arms fully extended, and reach the maximum possible distance by sliding the palms of the hands, one on top of the other, on the box [47].

The handgrip strength has been shown to be a valid test to measure musculoskeletal fitness in adolescents [48] (Figure 2). Considering previous research, the participants performed the test with the elbow fully extended, this being the most appropriate position to assess maximal strength in adolescents [49]. A Takei Tkk5401 digital handheld dynamometer (Takei Scientific Instruments, Tokyo, Japan) was used to measure the force produced.

The 20 m sprint indicated the minimum possible time taken by the subject to cover the indicated distance (Figure 2). For its performance, the participant stood statically on the initial line until the sprint began at maximum speed [50]. Single-beamed photocells (Polifemo Light, Microgate, Italy) located at hip height were used, the probability of being cut by the arms when running decreasing to 4%, compared to 60% probability when placed at chest height [51,52].

The countermovement jump (CMJ) test was used to evaluate the explosive power of the lower limbs through the height of the jump (Figure 2). This test consists of a vertical jump in which the participants had to stand in a standing position with their hands on their waists, flex their knees to a 90° position, and perform a full knee extension to reach the maximum possible height, keeping their hands on their hips and their trunk fully extended in the flight phase [53]. A force platform with a sampling frequency of 200 Hz (MuscleLab, Stathelle, Norway) was used to perform the test.

The 20 m shuttle run [54] is an incremental test with high validity and reliability for measuring cardiorespiratory fitness in adolescents [55] (Figure 2). The test ends when the participant reaches exhaustion or when he/she is not able to run the 20 m before the beep occurs. The speed at which the subject leaves the test is used to predict maximal oxygen consumption (VO2 max.) using the formula from Leger et al. [54].

### 2.4. Procedure

The physical fitness tests were carried out in the selected high schools, using the class hour belonging to the physical education session and taking advantage of the covered sports pavilions to reduce the polluting variables of the environment as much as possible. All tests were performed on the same day.

For data collection, a protocol was established in which the adolescents first completed questionnaires on their level of physical activity and nutritional habits. Subsequently, anthropometric measurements were taken to determine body composition by three accredited ISAK anthropometrists. Next, the sit-and-reach test was performed before the warm-up because previous research has reported the influence of warm-up on sit-and-reach performance in adolescents [56]. Once this test was completed, the execution of the hand grip strength, CMJ, and 20 m sprint tests were explained to the adolescents, familiarizing them for the correct execution of each test and participants completed a warm-up. The warm-up consisted of 5 minutes of progressive running and 10 minutes of joint mobility of the joints involved in the physical condition tests (ankles, knees, hips, wrists, and shoulders, mainly). A researcher supervised the warm-up and at the end of the warm-up told the participant which physical test to go to. Handgrip strength, the CMJ, and 20 m sprint tests took place randomly for each adolescent. After completion of the handgrip, CMJ, and 20 m sprint tests, the 20 m shuttle run test was performed. All fitness tests were performed twice for each adolescent, leaving two minutes between the two attempts of each test, and five minutes between tests, considering the best value reported, except for the sit-and-reach test and the 20 m run test, which were performed once. Four researchers with previous experience on the assessment of physical fitness tests oversaw the familiarization and assessment of these tests, with the same researcher being responsible for each test during all the measurements, in order to avoid inter-evaluator error in the assessments.

The order of the tests was selected according to the recommendations of the National Strength and Conditioning Association (NSCA), which bases its recommendation on the fatigue generated by the different tests, as well as the metabolic pathways required by each of them [57].

### 2.5. Data Analysis

The distribution of the data was initially evaluated using the Kolmogorov–Smirnov normality test, with all the variables having a normal distribution and allowing statistical analyses based on parametric tests. Descriptive statistics were used to find the mean values and standard deviation. A one-factor ANCOVA was performed to analyze the differences between physically active and sedentary adolescents, with gender, age, BMI, and biological maturation as covariates in the model. A binary logistic regression analysis was used to determine the primary outcomes for establishing differences between active and sedentary adolescents. A MANOVA analysis was performed to establish the differences based on the two independent variables physical activity practice and BMI between sedentary normal weight, sedentary over-weight, sedentary under-weight, active normal weight, active over-weight, and active under-weight adolescents in all analyzed variables. Bonferroni’s pairwise comparison was used for variables that were statistically significant. Partial eta squared (η^2^) was used to calculate the effect size and was defined as: small: ES ≥ 0.10; moderate: ES ≥ 0.30; large: ≥1.2; or very large: ES ≥ 2.0, with an error of *p* < 0.05 [58]. A value of *p* < 0.05 was set to determine statistical significance. The statistical analysis was performed with the SPSS statistical package (v. 25.0; SPSS Inc., Chicago, IL, USA).

## 3. Results

### 3.1. Intraclass Correlation Coefficients (ICC) and Coefficient of Variation (CV) in Fitness Tests

The intraclass correlation coefficients (ICC) and coefficient of variation (CV) were calculated for the fitness tests that were performed twice. The results were: handgrip strength right arm: ICC = 0.940, CV = 3.14%; handgrip strength left arm: ICC = 0.953, CV = 3.10%; CMJ: ICC = 0.892, CV = 3.35%; and 20 m sprint: ICC = 0.913, CV = 2.15%.

### 3.2. Differences between Active and Sedentary Adolescents

The results of the comparison between active and sedentary adolescents in the variables analyzed, and the main effect of the covariates, can be found in Table 1. The differences were significant between active and sedentary adolescents (*p* = 0.019–<0.001) in most of the anthropometric, AMD, and physical fitness variables, with a significant effect of the covariates gender, age, BMI, and biological maturation (*p* = 0.039 < 0.001) on the model. Specifically, according to the basic measurements, only height showed significant differences between active and sedentary adolescents (*p* = 0.019), with active adolescents being taller than sedentary adolescents, with a small effect size. Most of the anthropometric variables showed significant differences (*p* < 0.001), with fat mass and sum of three skinfolds being higher in sedentary adolescents, while muscle mass, corrected girths, and waist–hip ratio higher in active adolescents, with a small effect size. Additionally, the nutritional habits showed significant differences (*p* < 0.001), with AMD being higher in active adolescents, with a small effect size. Regarding physical fitness variables, significant differences were found in all variables (*p* < 0.001), with active adolescents showing the greatest cardiorespiratory fitness, strength, speed, and jump height, with a small effect size.

The covariate gender showed main effects in all variables (<0.001), except in BMI (*p* = 0.210) and hip girth (*p* = 0.691); the covariate age showed main effects in all variables (*p* < 0.001–0.002), except in the waist–height ratio (*p* = 0.247); the covariate BMI showed main effects in all variables (*p* < 0.001), except in the sit-and-reach test (*p* = 0.395); and the covariate biological maturation showed main effects in all variables (*p* < 0.001–0.039) except in body mass (*p* = 0.233), height (*p* = 0.068), and BMI (*p* = 0.101).

### 3.3. Binary Logistic Regression Analysis to Determine the Primary Outcome among the Significant Variables to Distinguish Active and Sedentary Groups

Table 2 shows the explanatory model for the level of physical activity. The results show that body mass (*p* = 0.017), BMI (*p* = 0.019), muscle mass (*p* = 0.018), hip girth (0.024), waist–hip ratio (*p* = 0.017), waist–height ratio (*p* = 0.050), AMD (*p* < 0.001), and VO2 max. (*p* < 0.001) can be considered as primary outcomes to distinguish between active and sedentary groups of adolescents.

### 3.4. Differences between Sedentary Adolescents According to Weight Status

Figure 3 and Table 3 show the differences between sedentary normal weight (*n* = 253), sedentary over-weight (*n* = 64), sedentary under-weight (*n* = 87), active normal weight (*n* = 212), active over-weight (*n* = 48), and active under-weight adolescents (*n* = 78) (*p* < 0.001–0.041). The pairwise comparison between sedentary normal weight–sedentary overweight–sedentary underweight showed significant differences in all variables analyzed except for height (*p* = 0.136–0.340), AMD (*p* = 0.316–0.599), and the sit-and-reach test (*p* = 0.173–0.475). Sedentary overweight adolescents showed higher values for all basic measurements and anthropometric variables than normal weight and underweight adolescents (*p* < 0.001). Regarding physical fitness variables, this group obtained the highest values in handgrip strength, but their performance was significantly lower in cardiorespiratory fitness, jump height, and 20 m sprint (*p* < 0.001–0.038).

Between sedentary normal weight and underweight adolescents, the normal weight adolescents presented higher values in body mass and anthropometric variables (*p* < 0.001). Significant differences were also observed in handgrip strength, with normal weight adolescents showing higher values (*p* < 0.001), but no significant differences were found in the rest of the physical fitness variables.

### 3.5. Differences between Active Adolescents According to Weight Status

Similarly, the differences between active normal weight–active overweight–active underweight adolescents were significant in all variables except for AMD and 20 m sprint. Significantly higher values were found in active overweight adolescents compared to active normal weight and active underweight in all anthropometric variables (*p* < 0.001–0.007). Regarding the physical fitness variables, active underweight adolescents showed significantly higher VO2 max values compared to active overweight ones (*p* = 0.024); handgrip strength was higher in active overweight than in active normal weight (*p* = 0.002–0.005) and active underweight (*p* < 0.001), as well as higher in active normal weight than in active underweight (*p* < 0.001). Lastly, active normal weight showed greater sit-and-reach distance than active underweight (*p* = 0.006), and greater jump height than active overweight (*p* = 0.013).

### 3.6. Differences between Active and Sedentary Adolescents with the Same Weight Status

When comparing the results obtained for sedentary normal weight and active normal weight, the differences were significant in all the anthropometric variables (*p* < 0.001–0.030), with sedentary adolescents showing higher values for fat mass and sum of three skinfolds, while active adolescents showed more muscle mass, corrected girths, waist girth, and waist–hip ratio. AMD was significantly higher in active adolescents (*p* < 0.001). With respect to the physical condition variables, the active adolescents showed higher performance in all the tests (*p* < 0.001–0.001), except for the sit-and-reach test, in which no significant differences were observed (*p* = 0.995).

The differences were also significant in sedentary and active overweight adolescents. In this case, sedentary adolescents had higher levels of fat mass and sum of three skinfolds, while active adolescents had more muscle mass and corrected thigh and calf girth (*p* < 0.001–0.034). Active overweight adolescents showed better performance in all physical fitness tests (*p* < 0.001–0.041), except for the sit-and-reach test, where no significant differences were found (*p* = 0.898).

Regarding active and sedentary underweight adolescents, the differences were significant in fat mass, muscle mass, and waist–hip ratio, with active adolescents having higher muscle mass and waist–hip ratio (*p* = 0.006–0.038). AMD and performance in the physical fitness tests were higher in the active adolescents (*p* < 0.001–0.039), except for the sit-and-reach test, where no significant differences were found (*p* = 0.262).

## 4. Discussion

The main objectives of the present investigation were (a) to establish the differences in body composition, physical performance, and AMD between active and sedentary adolescents; and (b) to analyze the differences between active and sedentary adolescents according to the “fat but fit” paradigm considering different weight statuses. Based on the results of previous research, the following hypotheses were put forward: (a) active adolescents will show better body composition, physical fitness, and AMD than sedentary adolescents; and (b) adolescents who show a higher level of physical activity will present higher physical fitness and better body composition, independently of their weight status. A significant difference was found between active and sedentary adolescents in most of the anthropometric, AMD, and physical fitness variables, with a significant effect of the covariates gender, age, BMI, and biological maturation on the model. The binary logistic regression analysis performed shows that the variables body mass, BMI, muscle mass, hip girth, waist-to-hip ratio, waist-to-height ratio, AMD, and VO2 max can be considered as primary outcomes to distinguish between active and sedentary groups of adolescents, and special attention should be paid to the differences in these variables. When comparing the differences between groups according to physical activity and BMI, it was found that, in general, at the same level of physical activity, adolescents with higher BMI showed higher values for anthropometric variables and worse values in the physical fitness tests. On the other hand, at the same BMI, sedentary adolescents generally showed higher values for anthropometric variables and worse values in the physical fitness tests and the AMD.

According to the first objective of the present study, to establish the differences in body composition, physical performance, and AMD between active and sedentary adolescents, the results obtained showed that active adolescents had lower values of fat mass and sum of skinfolds, and higher values of height, muscle mass, corrected girths, and waist–hip ratio. In fact, body mass and BMI, muscle mass, hip girth, waist-to-hip ratio, and waist-to-height ratio were found as primary outcomes to distinguish between active and sedentary groups. Previous studies have also found that active adolescents presented higher values of muscle mass and lower values of fat mass, as compared to sedentary adolescents [8,9,19], which could be because regular physical activity produces improvements in body composition during adolescence [59].

Surprisingly, the present study also found that active adolescents were taller than sedentary adolescents. Only one previous study has analyzed the difference in height between active and sedentary subjects and found no difference between them [60]. However, previous studies have shown that early maturers develop a series of physical and anthropometric capacities as a result of the maturational effect, including being taller than normal and late maturers during this period of growth [61,62]. This aspect could be a competitive advantage in sports involving strength, power, speed, agility, and endurance [63], i.e., most competitive sports played by adolescents, including basketball, football, and martial arts [64,65]. Also in those sports that are usually practiced in Physical Education sessions within the school environment, among which volleyball, basketball, handball, and football stand out [65,66]. The fact that early maturing adolescents have a competitive advantage over their peers, albeit temporary, could lead them to have a greater sense of their level of competence [67], this being a key aspect for the maintenance of an activity according to the self-determination theory [68]. Not surprisingly, previous studies have suggested that the adolescent’s perception of being unable to meet the demands of sport leads to withdrawal from sport [69]. This could lead to late maturers tending to abandon sport to a greater extent [70], which would explain the differences found in height between the active and sedentary adolescents in the present study.

However, the fact that significant differences between active and sedentary adolescents were found in the present study may be because adolescents who are taller in their age range could practice physical activity for a longer period of time than those who are shorter. This could be due to the fact that early maturation, the development of physical capacities, and height, are determining factors for sports performance [61,62], and could influence the level of sports participation, with taller adolescents maintaining their level of sports practice. However, this should be corroborated in future studies.

AMD was also shown to be higher in active adolescents and a primary outcome to distinguish between those who are active and sedentary. Previous studies have suggested that there may be a link between healthy physical activity habits and nutritional habits [7,71]. This could be because the establishment of a certain healthy lifestyle habit, such as regular physical activity, facilitates the adoption of other healthy lifestyle habits, such as maintaining a healthy diet [72].

Regarding the physical condition variables, upper and lower limb strength, sprint speed, and cardiorespiratory fitness, were higher in active adolescents, while the distance reached in the sit-and-reach test showed no significant differences with respect to sedentary adolescents. Scientific evidence has highlighted an association between higher physical activity practice and an increase in physical fitness levels [73,74]. This is evidence that, from an early age, being physically active is synonymous with less difficulty and effort in facing physical tasks [75]. More specifically, previous studies have already pointed out that there might be differences in all physical tests that depend on cardiorespiratory endurance, muscle strength, muscle endurance, muscle power, speed, balance, coordination, accuracy, and agility between active and sedentary subjects, but not in tests that depend on flexibility [9,60]. This could be because the practice of regular physical activity increases muscular fitness and cardiorespiratory fitness in adolescents [76,77], but flexibility requires a high volume of stretching for range of motion to improve, which is not common in most sports practices [78]. In addition, a significant finding of our study was that only cardiorespiratory fitness showed itself to be a primary outcome to distinguish between active and sedentary individuals. Cardiorespiratory fitness has been proposed by previous studies as the parameter of the physical condition variables most related to healthy lifestyle [79]. Therefore, it could be the most important physical condition parameter to differentiate between active and sedentary adolescents. After these promising results, questions remain for future studies, such as whether gender influences this or whether it would be the same in children.

The gender covariate showed main effects in all variables analyzed, except in BMI and hip girth. Previous studies have already pointed out that males tend to show lower values for anthropometric variables related to adiposity and higher values for variables related to muscle development [44] than females. It has been suggested that during this stage a sexual dimorphism occurs as a result of the differences in hormone production after puberty [80] which favors men to increase their muscle mass, while women have a greater tendency to accumulate adiposity [44,61,65]. Gender differences have also been found in adolescents, with males showing a higher level of physical fitness than females [81], which could be due to biological differences between them [44,61,65], but also to adaptive issues, as males are generally more active than females [81]. Gender differences are less clear with regard to dietary habits as, while previous studies have suggested that they may have similar AMD [82,83], other studies have indicated that dietary patterns may be different according to gender [84]. Therefore, these remaining questions need to be addressed in future studies.

The inclusion of the covariate age showed significant differences in all variables analyzed, except for the waist–height ratio. Previous studies have found that older adolescents presented higher scores in physical fitness tests, greater muscle development, greater AMD, and higher levels of fat mass than the youngest [85,86]. This is because adolescents are in the midst of a maturation process in which the production of growth hormone and sex steroid hormones increases with age [87], being related to the changes produced in muscle mass and fat mass [85], and physical capabilities that rely on strength, power, or endurance [65]. Regarding the waist–height ratio variable, it is limited by the height of the adolescents, and it increases throughout adolescence [86], with this being a possible explanation for the absence of significant differences when considering this covariate.

The BMI covariate also showed significant differences in all variables, except for the sit-and-reach score. These results are not surprising and are in agreement with those found in previous research in which adolescents with higher BMI scores had higher levels of fat mass and muscle mass [88], as well as worse performance in physical fitness tests [89]. This is a determinant during adolescence, because difficulties in physical performance and distorted body image hinder the relationship with peers [90]. The absence of significant differences in the sit-and-reach score is also in line with previous research [91] and could indicate that BMI is not a determining factor in the flexibility of adolescents.

On the inclusion of the covariate biological maturation, it was found that it had a significant effect on the differences found between the sedentary and active adolescents in all anthropometric and physical variables and AMD, except for the variables body mass, height, and BMI. Previous studies have already pointed to the influence of maturation on anthropometric and derived variables and physical performance among males [61,65]. More specifically, it has been found that adolescent boys whose maturational process is more advanced have a competitive advantage during the growth period in all physical condition variables that depend on the ability to produce strength and power [65,92], which could be due in part to their greater muscle mass as a result of hormonal changes in general, and specifically the increase in the amounts of testosterone that occurs during this stage [92,93]. Results are much weaker among adolescent females, although the trend appears to be similar [61,65]. Moreover, no previous studies have analyzed the interaction of physical activity and maturation on eating habits. This should be addressed in future research.

The second objective of the present investigation was to analyze the differences between active and sedentary adolescents according to the “fat but fit” paradigm considering different weight status. When comparing the groups of active and sedentary adolescents with normal weight, the differences were significant in the variables of fat and muscle mass, performance in physical fitness tests, and AMD, with the active adolescents obtaining greater values in all variables. Similar results were found when comparing active and sedentary overweight and underweight adolescents, except for AMD, in which overweight adolescents showed no differences according to the level of physical activity practiced. Paying special attention to the group of overweight/obese adolescents, previous studies have found that active adolescents show better physical performance and lower body fat than sedentary adolescents despite being overweight, giving rise to the so-called “fat but fit” paradigm [23]. However, the absence of differences in AMD between active and sedentary adolescents in the group of overweight adolescents could indicate that the practice of physical activity in this group is not as decisive for the adoption of healthy lifestyle habits as in the normal weight group. Previous studies have suggested that there is a relationship between physical exercise and AMD, but without addressing the issue from a “fat but fit” paradigm [35]. This is an issue that should be pursued in future studies. Therefore, on the basis of the results of the present investigation, a combined intervention of physical activity and healthy nutritional habits would be necessary in the overweight population if changes in adiposity accumulation are to occur.

Regarding the comparison of overweight, normal weight, and underweight sedentary subjects, as well as overweight, normal weight, and underweight active subjects, the results in the sedentary and active adolescent groups showed higher values in overweight adolescents in the anthropometric variables as compared to normal weight and underweight adolescents. In addition, overweight adolescents presented lower scores in physical fitness tests, except for hand grip strength, in both the active and sedentary adolescent groups, when compared to normal weight and underweight adolescents. Previous research has also shown that overweight adolescents showed lower scores in jump height, cardiorespiratory fitness, and speed tests, but higher scores in handgrip strength, than normo- and underweight adolescents [94,95]. An explanation could be that in physical tests in which body mass is mobilized, overweight and obese adolescents present limitations due to excess weight. However, the development of muscle strength in overweight subjects is greater than that of normal and underweight subjects, as they have higher levels of bone and muscle mass [96].

Notably, in underweight active adolescents, the handgrip strength score was significantly lower compared to overweight and normal weight adolescents, while VO2 max was significantly higher than that of overweight active subjects. Previous data has shown that underweight adolescents had better cardiorespiratory fitness than overweight adolescents [97], while other research has shown that underweight adolescents had worse cardiorespiratory fitness than normal weight adolescents [98]. Therefore, it seems evident that the physical condition of adolescents does not differ only when considering weight status, and that factors such as the type of sports practice or the frequency of training may have an influence. Future research is needed to further analyze the factors influencing the development of physical capacities during adolescence.

Regarding the practical implications of this research, the practice of physical activity during adolescence is essential for an adequate development of body composition, physical capacities, and AMD. In view of the current curriculum and the reduced hours of school physical activity in Spain, the main solution is to give more importance to the subject of physical education within the framework of compulsory education, by increasing the number of hours of physical education in the Spanish educational curriculum, but also by carrying out programs to raise awareness among all those involved in the educational process (teachers of other areas, teachers of the area itself, parents, students, etc.) of its importance and influence on the future health of adolescents. As a secondary solution, greater importance should be given to the promotion of out-of-school physical activity, increasing the weekly hours of practice, and facilitating access to practice by adolescents, encouraging them to be active for longer each day. These strategies should place special emphasis on overweight and obese adolescents, since they are less active, and the health benefits obtained would be similar to those of adolescents with a better weight status.

The present study is not free of limitations. Since this was a cross-sectional design, it was not possible to establish a causal relationship between the variables analyzed. The sample was selected by convenience in the educational centers that could be accessed. Another possible limitation could be the selection of the questionnaire chosen to classify the subjects into active and sedentary, since, although it had been previously validated in a sample of Spanish adolescents, showing adequate validity and reliability for measuring physical activity [28], the results could be influenced by the measurement instrument, being a possible limitation of this study. For this reason, it would be interesting to support the results already obtained by using better measuring instruments for a direct quantification of the PA levels. Further research is needed in order to corroborate the main findings. Furthermore, the fact that the adolescents were classified as underweight, normal weight, and overweight/obese according to the World Health Organization classification, was not considered a limitation, but it is an aspect to be considered in future research, although there are other classifications that could modify the results obtained. Finally, BMI does not allow differentiation between adipose and muscle components [99], and future research will need to analyze this “fat but fit” phenomenon by using variables exclusively related to adiposity, such as the skinfold sums, the adipose/fat mass or the adipose/fat percentage, to classify adolescents as “fat”.

## 5. Conclusions

Considering the results obtained in the present research, it can be concluded that active adolescents have lower fat mass, higher values of muscle mass and height, greater AMD, greater strength, speed, and cardiorespiratory fitness than sedentary adolescents. Body mass, BMI, muscle mass, hip girth, waist-to-hip ratio, waist-to-height ratio, AMD, and VO2 max were considered as primary outcomes to distinguish between active and sedentary groups of adolescents The comparison between adolescents with different weight status showed higher results in anthropometric variables and lower performance in physical fitness tests, except for handgrip strength, in overweight or obese adolescents, regardless of whether they are active or sedentary. In addition, the more active adolescents within the same weight status group had greater muscle mass, physical performance, and AMD diet than the more sedentary adolescents, in accordance with the “fat but fit” paradigm.

## Figures and Tables

**Figure 1 ijerph-19-10797-f001:**
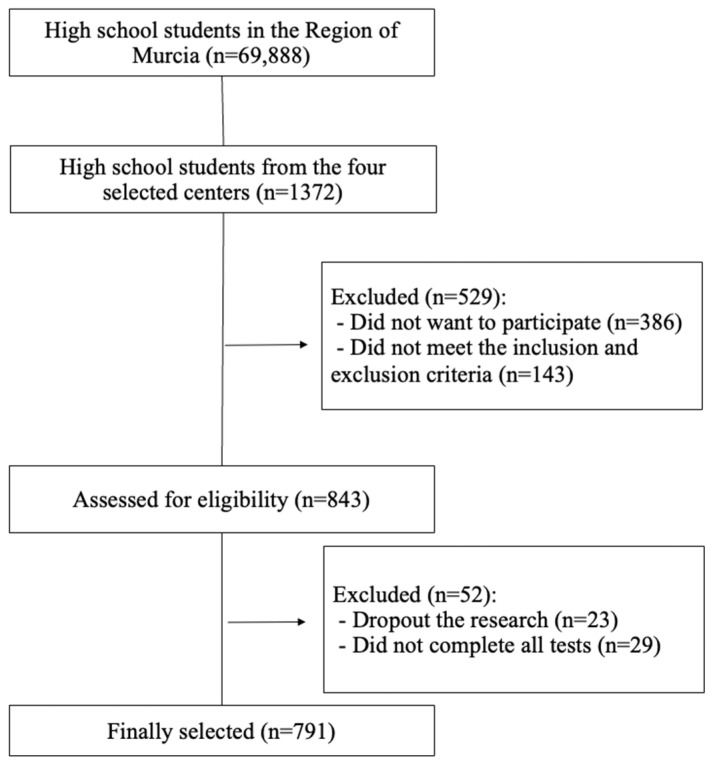
Sample selection flow chart.

**Figure 2 ijerph-19-10797-f002:**
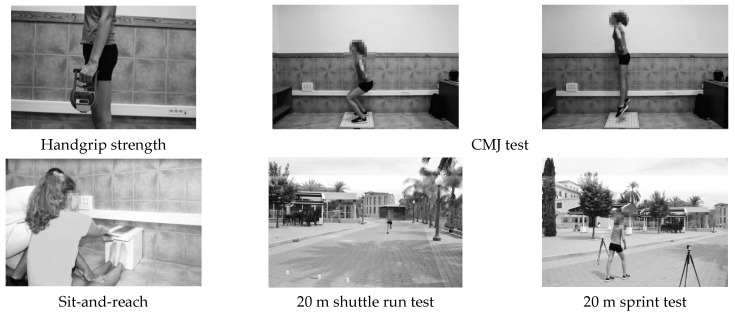
Physical performance tests.

**Figure 3 ijerph-19-10797-f003:**
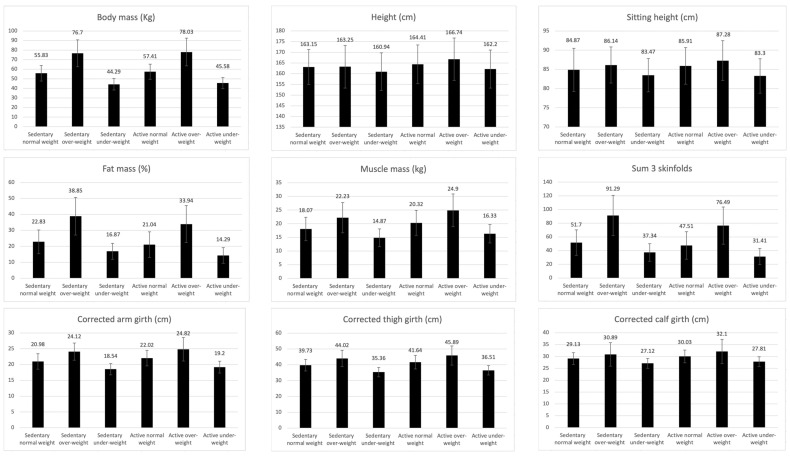
Descriptive values of anthropometric, nutritional, and physical fitness variables of active and sedentary adolescents according to their BMI.

**Table 1 ijerph-19-10797-t001:** Differences in anthropometric, nutritional habits, and physical fitness, between active and sedentary adolescents.

	Descriptive Statistics (M ± SD)		Active vs. Sedentary				Active vs. Sedentary * Gender			Active vs. Sedentary * Age			Active vs. Sedentary * BMI			Active vs. Sedentary * Biological Maturation		
	Active (*n* = 348)	Sedentary (*n* = 443)	F; *p*	Mean Diff.	95% CI	Effect Size (η^2^)	F	*p*	Effect Size (η^2^)	F	*p*	Effect Size (η^2^)	F	*p*	Effect Size (η^2^)	F	*p*	Effect Size (η^2^)
Body mass (Kg)	57.70 ± 13.04	56.77 ± 13.32	0.967; *p* = 0.326	0.956	−0.953; 2.864	0.001	26.954	<0.001	0.068	73.228	<0.001	0.165	1133.628	<0.001	0.754	1.461	0.233	0.004
Height (cm)	164.33 ± 9.18	162.76 ± 8.73	5.547; *p* = 0.019	1.545	0.257; 2.832	0.007	76.630	<0.001	0.165	109.171	<0.001	0.228	11.684	<0.001	0.031	2.702	0.068	0.007
Sitting height (cm)	85.52 ± 4.91	84.80 ± 5.21	3.786; *p* = 0.052	0.734	−0.007; 1.474	0.005	18.754	<0.001	0.048	105.134	<0.001	0.222	23.915	<0.001	0.061	31.681	<0.001	0.078
BMI (kg/m^2^)	21.30 ± 3.70	21.33 ± 4.06	0.001; *p* = 0.980	−0.007	−0.574; 0.559	0.001	1.566	0.210	0.004	24.320	<0.001	0.059	-	-	-	2.299	0.101	0.006
Fat mass (%)	21.22 ± 9.96	24.14 ± 10.39	13.519; *p* < 0.001	−2.768	−4.246; −1.290	0.018	27.233	<0.001	0.068	7.180	0.001	0.019	420.481	<0.001	0.532	27.853	<0.001	0.070
Muscle mass (kg)	20.11 ± 5.22	18.06 ± 4.80	29.505; *p* < 0.001	2.010	1.284; 2.737	0.038	245.187	<0.001	0.397	98.176	<0.001	0.210	179.129	<0.001	0.327	40.475	<0.001	0.098
Sum 3 skinfolds	47.67 ± 23.80	55.03 ± 25.81	14.400; *p* < 0.001	−6.969	−10.574; −3.364	0.019	35.162	<0.001	0.086	7.823	<0.001	0.021	390.044	<0.001	0.514	29.562	<0.001	0.074
Corrected arm girth (cm)	21.79 ± 3.05	20.96 ± 2.89	13.850; *p* < 0.001	0.819	0.387; 1.250	0.018	81.292	<0.001	0.179	80.181	<0.001	0.178	272.147	<0.001	0.424	13.971	<0.001	0.036
Corrected thigh girth (cm)	41.12 ± 5.21	39.52 ± 4.56	19.682; *p* < 0.001	1.598	0.891; 2.305	0.026	61.327	<0.001	0.142	72.638	<0.001	0.164	249.192	<0.001	0.403	11.850	<0.001	0.031
Corrected calf girth (cm)	29.87 ± 3.30	28.97 ± 3.13	12.340; *p* < 0.001	0.835	0.369; 1.302	0.016	47.761	<0.001	0.114	69.306	<0.001	0.158	82.276	<0.001	0.182	9.412	<0.001	0.025
Waist girth (cm)	70.19 ± 8.40	69.38 ± 9.06	1.534; *p* = 0.216	0.803	−0.470; 2.076	0.002	49.803	<0.001	0.118	27.476	<0.001	0.069	1258.918	<0.001	0.773	3.721	0.025	0.010
Hip girth (cm)	90.50 ± 9.05	91.12 ± 9.26	0.685; *p* = 0.408	−0.558	−1.881; 0.765	0.001	0.370	0.691	0.001	73.275	<0.001	0.165	1459.358	<0.001	0.798	11.716	<0.001	0.031
Waist–hip ratio	0.78 ± 0.60	0.76 ± 0.51	13.239; *p* < 0.001	0.015	0.007; 0.023	0.018	227.949	<0.001	0.380	11.213	<0.001	0.029	29.043	<0.001	0.073	87.132	<0.001	0.190
Waist–height ratio	0.43 ± 0.46	0.43 ± 0.53	0.060; *p* = 0.807	0.001	−0.006; 0.008	0.001	12.746	<0.001	0.033	1.401	0.247	0.004	1204.244	<0.001	0.765	3.250	0.039	0.009
AMD	6.97 ± 2.37	6.15 ± 2.46	21.885; *p* < 0.001	0.818	0.475; 1.161	0.029	12.177	<0.001	0.032	12.546	<0.001	0.033	11.292	<0.001	0.030	14.621	<0.001	0.038
VO2 max. (mL/kg/min)	41.81 ± 5.59	37.86 ± 5.19	100.51; *p* < 0.001	3.967	3.190; 4.744	0.120	138.648	<0.001	0.272	59.119	<0.001	0.138	70.858	<0.001	0.161	69.683	<0.001	0.158
Handgrip right arm (kg)	28.15 ± 8.78	25.64 ± 7.70	18.554; *p* < 0.001	2.603	1.417; 3.789	0.024	105.525	<0.001	0.221	127.609	<0.001	0.257	53.076	<0.001	0.126	24.235	<0.001	0.061
Handgrip left arm (kg)	26.09 ± 7.73	23.85 ± 7.27	17.497; *p* < 0.001	2.303	1.222; 3.383	0.023	119.805	<0.001	0.244	111.783	<0.001	0.232	48.736	<0.001	0.117	26.244	<0.001	0.066
Sit-and-reach (cm)	15.69 ± 8.98	16.03 ± 8.56	0.350; *p* = 0.554	−0.382	−1.650; 0.886	0.001	60.494	<0.001	0.140	23.031	<0.001	0.059	0.929	0.395	0.003	10.127	<0.001	0.027
CMJ (cm)	25.26 ± 7.12	22.20 ± 6.74	37.472; *p* < 0.001	3.073	2.088; 4.059	0.048	78.278	<0.001	0.174	51.549	<0.001	0.122	28.046	<0.001	0.071	53.184	<0.001	0.125
20 m sprint (s)	3.80 ± 0.46	4.03 ± 0.57	35.442; *p* < 0.001	−0.230	−0.305; −0.154	0.046	78.655	<0.001	0.175	57.034	<0.001	0.134	20.885	<0.001	0.053	55.954	<0.001	0.131

BMI: body mass index; AMD: Adhesion to Mediterranean Diet; VO2 max: maximum oxygen consumption; CMJ: countermovement jump; *: Indicates that the model analyzed is composed of both variables.

**Table 2 ijerph-19-10797-t002:** Binary logistic regression analysis of the analyzed variables to explain the differences between active and sedentary adolescents.

Variable	B	Standard Error	Sig.	Exp (B) Odds Ratios	95% CI
Body mass (Kg)	−0.194	0.081	0.017	0.823	0.702; 0.966
Height (cm)	−0.146	0.085	0.085	0.864	0.732; 1.020
Sitting height (cm)	0.056	0.039	0.151	1.057	0.980; 1.140
BMI (kg/m^2^)	0.533	0.228	0.019	1.704	1.090; 2.664
Fat mass (%)	0.014	0.050	0.785	1.014	0.919; 1.118
Muscle mass (kg)	0.242	0.103	0.018	1.274	1.042; 1.558
Sum 3 skinfolds	−0.003	0.021	0.903	0.997	0.958; 1.039
Corrected arm girth (cm)	−0.048	0.081	0.551	0.953	0.814; 1.116
Corrected thigh girth (cm)	−0.075	0.059	0.207	0.928	0.826; 1.042
Corrected calf girth (cm)	−0.049	0.049	0.317	0.952	0.866; 1.048
Waist girth (cm)	0.054	0.283	0.848	1.056	0.606; 1.839
Hip girth (cm)	0.342	0.152	0.024	1.407	1.046; 1.895
Waist–hip ratio	41.361	17.368	0.017	0.001	0.002; 0.765
Waist–height ratio	−83.247	42.382	0.050	0.001	0.001; 0.836
AMD	0.148	0.036	<0.001	1.160	1.081; 1.244
VO2 max. (mL/kg/min)	0.113	0.020	<0.001	1.120	1.078; 1.164
Handgrip right arm (kg)	0.002	0.026	0.935	1.002	0.952; 1.055
Handgrip left arm (kg)	−0.030	0.028	0.286	0.970	0.918; 1.026
Sit-and-reach (cm)	0.008	0.011	0.469	1.008	0.987; 1.029
CMJ (cm)	0.019	0.016	0.236	1.019	0.988; 1.052
20 m sprint (s)	−0.157	0.218	0.470	0.854	0.558; 1.309

BMI: body mass index; AMD: Adhesion to Mediterranean Diet; VO2 max: maximum oxygen consumption; CMJ: countermovement jump.

**Table 3 ijerph-19-10797-t003:** Bonferroni post hoc analysis of anthropometric, nutritional, and physical fitness variables of active and sedentary adolescents according to their BMI.

Variable	Comparison Groups	Mean Differences	95% CI	*p*	Effect Size (η^2^)
Body mass (Kg)	Sedentary normal weight–sedentary overweight	−20.873	−23.858; −17.889	<0.001	0.402
	Sedentary normal weight–sedentary underweight	11.533	8.882; 14.185	<0.001	0.402
	Sedentary overweight–sedentary underweight	32.407	28.894; 35.920	<0.001	0.402
	Active normal weight–active overweight	−20.620	−24.029; −17.210	<0.001	0.350
	Active normal weight–active underweight	11.834	9.009; 14.659	<0.001	0.350
	Active overweight–active underweight	32.454	28.540; 36.367	<0.001	0.350
Height (cm)	Active overweight–sedentary overweight	3.488	0.172; 6.804	0.039	0.001
	Active overweight–active underweight	4.541	0.647; 8.435	0.016	0.011
Sitting height (cm)	Active normal weight–sedentary normal weight	1.043	0.126; 1.960	0.026	0.007
	Sedentary overweight–sedentary underweight	2.668	0.685; 4.651	0.004	0.014
	Active normal weight–active underweight	2.609	1.015; 4.204	<0.001	0.030
	Active overweight–active underweight	3.974	1.765; 6.183	<0.001	0.030
Fat mass (%)	Active normal weight–sedentary normal weight	−1.786	3.243; −0.328	0.016	0.008
	Active overweight–sedentary overweight	−4.909	−7.898; −1.920	0.001	0.014
	Active underweight–sedentary underweight	−2.582	−5.023; −0.141	0.038	0.006
	Sedentary normal weight–sedentary overweight	−16.021	−18.698; −13.344	<0.001	0.287
	Sedentary normal weight–sedentary underweight	5.953	3.575; 8.330	<0.001	0.287
	Sedentary overweight–sedentary underweight	21.973	18.823; 25.124	<0.001	0.287
	Active normal weight–active overweight	−12.898	−15.956; −9.840	<0.001	0.197
	Active normal weight–active underweight	6.749	4.215; 9.282	<0.001	0.197
	Active overweight–active underweight	19.647	16.137; 23.156	<0.001	0.197
Muscle mass (kg)	Active normal weight–sedentary normal weight	2.251	1.441; 3.060	<0.001	0.039
	Active overweight–sedentary overweight	2.662	1.002; 4.321	0.002	0.013
	Active underweight–sedentary underweight	1.465	0.109; 2.820	0.034	0.006
	Sedentary normal weight–sedentary overweight	−4.166	−5.653; −2.680	<0.001	0.122
	Sedentary normal weight–sedentary underweight	3.199	1.879; 4.520	<0.001	0.122
	Sedentary overweight–sedentary underweight	7.365	5.616; 9.115	<0.001	0.122
	Active normal weight–active overweight	−4.577	−6.275; −2.879	<0.001	0.133
	Active normal weight–active underweight	3.986	2.579; 5.393	<0.001	0.133
	Active overweight–active underweight	8.563	6.614; 10.511	<0.001	0.133
Sum 3 skinfolds	Active normal weight–sedentary normal weight	−4.189	−7.784; −0.594	0.022	0.007
	Active overweight–sedentary overweight	−14.808	−22.180; −7.436	<0.001	0.021
	Sedentary normal weight–sedentary overweight	−39.596	−46.198; −32.993	<0.001	0.286
	Sedentary normal weight–sedentary underweight	14.361	8.496; 20.226	<0.001	0.286
	Sedentary overweight–sedentary underweight	53.957	46.186; 61.728	<0.001	0.286
	Active normal weight–active overweight	−28.977	−36.519; −21.434	<0.001	0.175
	Active normal weight–active underweight	16.099	9.850; 22.349	<0.001	0.175
	Active overweight–active underweight	45.076	36.419; 53.733	<0.001	0.175
Corrected arm girth (cm)	Active normal weight–sedentary normal weight	1.044	0.594; 1.495	<0.001	0.027
	Sedentary normal weight–sedentary overweight	−3.142	−3.969; −2.315	<0.001	0.204
	Sedentary normal weight–sedentary underweight	2.434	1.699; 3.169	<0.001	0.204
	Sedentary overweight–sedentary underweight	5.576	4.602; 6.550	<0.001	0.204
	Active normal weight–active overweight	−2.799	−3.744; −1.854	<0.001	0.179
	Active normal weight–active underweight	2.814	2.031; 3.597	<0.001	0.179
	Active overweight–active underweight	5.613	4.528; 6.698	<0.001	0.179
Corrected thigh girth (cm)	Active normal weight–sedentary normal weight	1.920	1.180; 2.659	<0.001	0.034
	Active overweight–sedentary overweight	1.878	0.361; 3.394	0.015	0.008
	Sedentary normal weight–sedentary overweight	−4.292	−5.650; −2.933	<0.001	0.189
	Sedentary normal weight—sedentary underweight	4.366	3.160; 5.573	<0.001	0.189
	Sedentary overweight—sedentary underweight	8.658	7.060; 10.257	<0.001	0.189
	Active normal weight–active overweight	−4.249	−5.801; −2.698	<0.001	0.189
	Active normal weight–active underweight	5.138	3.853; 6.424	<0.001	0.189
	Active overweight–active underweight	9.388	7.607; 11.169	<0.001	0.189
Corrected calf girth (cm)	Active normal weight–sedentary normal weight	0.899	0.352; 1.446	0.001	0.014
	Active overweight–sedentary overweight	1.213	0.091; 2.334	0.034	0.006
	Sedentary normal weight–sedentary overweight	−1.757	−2.762; −0.753	<0.001	0.076
	Sedentary normal weight–sedentary underweight	2.014	1–122; 2.907	<0.001	0.076
	Sedentary overweight–sedentary underweight	3.772	2.590; 4.954	<0.001	0.076
	Active normal weight–active overweight	−2.072	−3.219; −0.924	<0.001	0.080
	Active normal weight–active underweight	2.223	1.272; 3.173	<0.001	0.080
	Active overweight–active underweight	4.294	2.977; 5.611	<0.001	0.080
Waist girth (cm)	Active normal weight–sedentary normal weight	1.153	0.113; 2.193	0.030	0.006
	Sedentary normal weight–sedentary overweight	−15.773	−17.683; −13.863	<0.001	0.451
	Sedentary normal weight–sedentary underweight	6.860	5.163; 8.557	<0.001	0.451
	Sedentary overweight–sedentary underweight	22.633	20.385; 24.881	<0.001	0.451
	Active normal weight–active overweight	−15.033	−17.215; −12.851	<0.001	0.367
	Active normal weight–active underweight	6.341	4.533; 8.148	<0.001	0.367
	Active overweight–active underweight	21.374	18.869; 23.878	<0.001	0.367
Hip girth (cm)	Sedentary normal weight–sedentary overweight	−14.500	−16.435; −12.566	<0.001	0.465
	Sedentary normal weight–sedentary underweight	9.476	7.758; 11.194	<0.001	0.465
	Sedentary overweight–sedentary underweight	23.976	21.700; 26.253	<0.001	0.465
	Active normal weight–active overweight	−13.601	−15.811; −11.391	<0.001	0.399
	Active normal weight–active underweight	9.700	7.870; 11.531	<0.001	0.399
	Active overweight–active underweight	23.301	20.765; 25.837	<0.001	0.399
Waist–hip ratio	Active normal weight–sedentary normal weight	0.014	0.004; 0.024	0.005	0.010
	Active underweight–sedentary underweight	0.023	0.007; 0.039	0.006	0.010
	Sedentary normal weight–sedentary overweight	−0.046	−0.064; −0.028	<0.001	0.049
	Sedentary overweight–sedentary underweight	0.041	0.020; 0.063	<0.001	0.049
	Active normal weight–active overweight	−0.043	−0.064; −0.023	<0.001	0.034
	Active overweight–active underweight	0.030	0.006; 0.053	0.007	0.034
Waist–height ratio	Sedentary normal weight–sedentary overweight	−0.097	−0.107; −0.086	<0.001	0.486
	Sedentary normal weight–sedentary underweight	0.037	0.027; 0.046	<0.001	0.486
	Sedentary overweight–sedentary underweight	0.133	0.121; 0.146	<0.001	0.486
	Active normal weight–active overweight	−0.084	−0.096; −0.072	<0.001	0.365
	Active normal weight–active underweight	0.033	0.023; 0.043	<0.001	0.365
	Active overweight–active underweight	0.118	0.104; 0.131	<0.001	0.365
AMD	Active normal weight–sedentary normal weight	0.785	0.351; 1.219	<0.001	0.017
	Active underweight–sedentary underweight	1.124	0.397; 1.850	0.002	0.012
VO2 max. (mL/kg/min)	Active normal weight–sedentary normal weight	3.906	2.941; 4.872	<0.001	0.079
	Active overweight–sedentary overweight	4.713	2.733; 6.694	<0.001	0.029
	Active underweight–sedentary underweight	3.401	1.784; 5.018	<0.001	0.023
	Sedentary normal weight–sedentary overweight	2.656	0.882; 4.430	0.001	0.027
	Sedentary overweight–sedentary underweight	−3.888	−5.976; −1.800	<0.001	0.027
	Active overweight–active underweight	−2.575	−4.901; −0.250	0.024	0.010
Handgrip right arm (kg)	Active normal weight–sedentary normal weight	2.609	1.180; 4.038	<0.001	0.017
	Active overweight–sedentary overweight	3.587	0.657; 6.517	0.016	0.008
	Active underweight–sedentary underweight	2.515	0.122; 4.907	0.039	0.006
	Sedentary normal weight–sedentary overweight	−3.315	−5.939; −0.690	0.008	0.050
	Sedentary normal weight–sedentary underweight	4.443	2.112; 6.774	<0.001	0.050
	Sedentary overweight–sedentary underweight	7.757	4.669; 10.846	<0.001	0.050
	Active normal weight–active overweight	−4.293	−7.291; −1.295	0.002	0.051
	Active normal weight–active underweight	4.537	2.053; 7.021	<0.001	0.051
	Active overweight–active underweight	8.830	5.389; 12.270	<0.001	0.051
Handgrip left arm (kg)	Active normal weight–sedentary normal weight	2.149	0.842; 3.456	0.001	0.014
	Active overweight–sedentary overweight	2.802	0.121; 5.483	0.041	0.006
	Active underweight–sedentary underweight	2.909	0.720; 5.099	0.009	0.009
	Sedentary normal weight–sedentary overweight	−2.988	−5.390; −0.587	0.009	0.052
	Sedentary normal weight–sedentary underweight	4.245	2.112; 6.378	<0.001	0.052
	Sedentary overweight–sedentary underweight	7.233	4.407; 10.060	<0.001	0.052
	Active normal weight–active overweight	−3.641	−6.384; −0.898	0.005	0.040
	Active normal weight–active underweight	3.485	1.212; 5.757	0.001	0.040
	Active overweight–active underweight	7.126	3.977; 10.274	<0.001	0.040
Sit-and-reach (cm)	Active normal weight–active underweight	3.584	0.821; 6.348	0.006	0.014
CMJ (cm)	Active normal weight–sedentary normal weight	2.892	1.663; 4.120	<0.001	0.028
	Active overweight–sedentary overweight	3.548	1.028; 6.067	0.006	0.010
	Active underweight–sedentary underweight	3.068	1.011; 5.126	0.004	0.012
	Sedentary normal weight–sedentary overweight	3.726	1.469; 5.982	<0.001	0.021
	Sedentary overweight–sedentary underweight	−2.853	−5.509; −0.197	0.030	0.021
	Active normal weight–active overweight	3.070	0.492; 5.648	0.013	0.011
20 m sprint (s)	Active normal weight–sedentary normal weight	−0.227	−0.323; −0.132	<0.001	0.029
	Active overweight–sedentary overweight	−0.287	−0.482; −0.091	0.004	0.011
	Active underweight–sedentary underweight	−0.191	−0.350; −0.031	0.019	0.007
	Sedentary normal weight–sedentary overweight	−0.182	−0.357; −0.007	0.038	0.008

BMI: body mass index; APHV: age at peak height velocity; AMD: Adhesion to Mediterranean Diet; VO2 max: maximum oxygen consumption; CMJ: countermovement jump.

## Data Availability

The data presented in this study are available on request from the corresponding author. The data are not publicly available due to their containing information that could compromise the privacy of research participants but are available from the corresponding author on reasonable request.

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
