# Peer review of "Physical, Psychological, and Body Composition Differences between Active and Sedentary Adolescents According to the “Fat but Fit” Paradigm"

_ijerph, 2022, doi:10.3390/ijerph191710797_

Round 1

Reviewer 1 Report

Major comments:

This study confirmed which assessments are useful to distinguish level of physical performances in the adolescent population. This study has a potential novelty, however, it is required several major revisions.

Firstly, the authors need to explain what the problem in this population is, why they do it, and how to solve the research hypotheses in the introduction section. Most explanation was jumped to other contents, which may be difficult to understand the detailed background of this study.

In addition, the authors demonstrated several variables were different between groups, but I am not sure what is the major outcome in this study. For instance, the authors explained several variables in the discussion section, but honestly, it makes me question "So what's your point?". I recommend performing additional statistics which variables are primary outcome among the significant variables to distinguish active and sedentary groups (e.g., binary logistic regression analysis).

Finally, the provided tables are containing too much information, as well as poor readability. I highly recommend separating the tables and utilizing figures to easily describe your research protocols and results. The specific comments are as follows:

Specific comments:

Introduction section

First paragraph: I understand what the authors mean. However, please add detailed information on what is the problem if the children or adolescent populations are not considered these physical activities and diet behaviors; and why they do not prefer those things.

Lines 53-54: Please clarify what is the specific information and required. In other words, please summarize the gaps between previous studies and your study.

Lines 56-57: Likewise, the authors need to explain what kinds of assessments are useful to distinguish the essential level of physical activities.

Lines 58-60: If you want to emphasize the necessity of this study, please review the detailed limitations of the previous studies, and suggest clearly based on the gaps.

Lines 63-64: Please add the hypotheses at the end of the introduction contents.

Methods section

Participants section: Please clarify and state if you received a consent agreement by the participants and their protectors.

And also, I highly recommend adding a flowchart for recruitment.

Lines 93-98: I recommend adding the range of each item and composite scores.

Line 99: Please state the full term of this abbreviation for the first time (KIDMED).

Line 106: Likewise, please mention here for the full term.

Lines 111-115: Please clarify how many researchers were considered to maintain reliability for the skin folder assessments and others.

Lines 127-130: Please add proper references.

Physical fitness test section: I recommend adding figures to describe each physical performance task.

Also, the authors should mention why these kinds of physical performance assessment tests are required to assess the adolescent population in the introduction section.

Lines 192-194: I am wondering why did you perform the MANOVA analysis? As far as I know, the MANOVA analysis is usually performed if the independent variables are at least 3 (e.g., groups, gender, visit time, etc). Please clarify these contents.

Results section: Please mention the correct effect sizes in each p-value instead of the subjective statement.

Table 1: Please clarify the gender distribution in each group.

Table 2: What is the meaning of these symbols? Besides, I highly recommend separating several figures or tables, because this table has too much information, as well as poor readability.

Besides, I recommend please remove the underline between variables in each table.

Discussion section: Before beginning to the discussion, I highly recommend summarizing the aim and hypothesis of this study, and the primary outcomes in the first paragraph.

Lines 293-296: This mention could be supported by the gender distribution. As you know, gender is also related to body height. As I mentioned above, the authors should clarify this issue in this manuscript.

Conclusion section.

As I mentioned above, please summarize which variables are primary to distinguish between active and sedentary adolescent populations performing additional statistics. It could be beneficial to provide primary outcome, as well as essential information according to the sub-categories (demographics, questionnaires, physical fitness tests, etc.).

Author Response

This study confirmed which assessments are useful to distinguish level of physical performances in the adolescent population. This study has a potential novelty, however, it is required several major revisions.

Firstly, the authors need to explain what the problem in this population is, why they do it, and how to solve the research hypotheses in the introduction section. Most explanation was jumped to other contents, which may be difficult to understand the detailed background of this study.

In addition, the authors demonstrated several variables were different between groups, but I am not sure what is the major outcome in this study. For instance, the authors explained several variables in the discussion section, but honestly, it makes me question "So what's your point?". I recommend performing additional statistics which variables are primary outcome among the significant variables to distinguish active and sedentary groups (e.g., binary logistic regression analysis).

Finally, the provided tables are containing too much information, as well as poor readability. I highly recommend separating the tables and utilizing figures to easily describe your research protocols and results.

+ Dear reviewer, thank you very much for your input. We will address each of them to improve the manuscript according to your indications.

The specific comments are as follows: 

Introduction section

First paragraph: I understand what the authors mean. However, please add detailed information on what is the problem if the children or adolescent populations are not considered these physical activities and diet behaviors; and why they do not prefer those things.

+ Thank you for your input. Information on the disadvantages of not adopting the recommendations has been included, as well as a paragraph with the reasons for not doing so.

Lines 53-54: Please clarify what is the specific information and required. In other words, please summarize the gaps between previous studies and your study.

+ Thank you very much for your comment. The gaps about the fat but fit paradigm and the contribution of the present research to it have been expanded.

Lines 56-57: Likewise, the authors need to explain what kinds of assessments are useful to distinguish the essential level of physical activities.

+ Thank you for your input. Information on the most commonly used ways of assessing the level of physical activity has been included.

Lines 58-60: If you want to emphasize the necessity of this study, please review the detailed limitations of the previous studies, and suggest clearly based on the gaps.

+Thank you for your comments. More information has been included about the gaps in the previous literature and the importance of this article in filling those gaps.

Lines 63-64: Please add the hypotheses at the end of the introduction contents.

+ Thank you for your input, two hypotheses have been included to give consistency to the introduction.

Methods section

Participants section: Please clarify and state if you received a consent agreement by the participants and their protectors. And also, I highly recommend adding a flowchart for recruitment.

+ Thank you very much. Done.

Lines 93-98: I recommend adding the range of each item and composite scores.

+ Thank you very much. Done.

Line 99: Please state the full term of this abbreviation for the first time (KIDMED).

+ Thank you very much. Done.

Line 106: Likewise, please mention here for the full term.

+ Thank you very much. Done.

Lines 111-115: Please clarify how many researchers were considered to maintain reliability for the skin folder assessments and others.

+ Thank you very much for your comment. It has been specified in the text that three anthropometrists taken all the measures and that all measurements corresponding to each subject were made by the same anthropometrist.

Lines 127-130: Please add proper references.

+ Thank you very much. Done.

Physical fitness test section: I recommend adding figures to describe each physical performance task. Also, the authors should mention why these kinds of physical performance assessment tests are required to assess the adolescent population in the introduction section.

+ Thank you very much for your comments. We include a paragraph in the introduction section and the pictures of each test. At this time, we are unable to provide higher quality images, as these are the ones we made during the investigation and our laboratory is closed for holidays. However, when we return, we will take professional photos and replace them.

Lines 192-194: I am wondering why did you perform the MANOVA analysis? As far as I know, the MANOVA analysis is usually performed if the independent variables are at least 3 (e.g., groups, gender, visit time, etc). Please clarify these contents.

+ Thank you very much for your comment. According to several statistical books consulted, among them Sánchez-Zuriaga, D. (2010). Estadística aplicada a la fisioterapia, las ciencias del deporte y la biomecánica. Valencia: CEU Ediciones or Navidi, W. (2006). Statistics for engineers and scientists. New York: McGraw-Hill, as we have more than one independent variable (in this case two: level of physical activity and BMI) and three or more groups (in this case we have six groups), a MANOVA factorial model has to be performed, as this allows to study the significance and nature of the differences between each pair of groups of subjects. We have clarified this in the section on statistical analysis.

Results section: Please mention the correct effect sizes in each p-value instead of the subjective statement.

+ Thank you for your input. The effect size of each comparison pair has been included in Table 4.

Table 1: Please clarify the gender distribution in each group.

+ Thank you very much. The n of each gender has been added in the “participants” section. Furthermore, Table 1 has been completed by introducing gender as a covariate.

Table 2: What is the meaning of these symbols? Besides, I highly recommend separating several figures or tables, because this table has too much information, as well as poor readability.

+ Thank you for this excellent comment. We have deleted in Table 3 all the symbols which referred to post hoc analysis, and we have created a new Table (Table 4) with this information, including the p values and effect sizes of all the variables that showed differences between groups in order to improve the readability of the information.

Besides, I recommend please remove the underline between variables in each table.

+ Thank you very much. Done.

Discussion section: Before beginning to the discussion, I highly recommend summarizing the aim and hypothesis of this study, and the primary outcomes in the first paragraph.

+ Thank you for your input. A first introductory paragraph has been included in the discussion.

Lines 293-296: This mention could be supported by the gender distribution. As you know, gender is also related to body height. As I mentioned above, the authors should clarify this issue in this manuscript.

+  Thank you very much for your great contribution. Gender has been included as a covariate to understand if it affected the model. The results found have been discussed. In addition, we believe that the analysis of gender differences requires further interpretation and has been proposed as a future line of research to be addressed.

Conclusion section.

As I mentioned above, please summarize which variables are primary to distinguish between active and sedentary adolescent populations performing additional statistics. It could be beneficial to provide primary outcome, as well as essential information according to the sub-categories (demographics, questionnaires, physical fitness tests, etc.).

+ Thank you for this valuable comment. Additional statistical analysis has been performed in order to detect these primary outcomes. These results have been included in Table 2. Furthermore, the findings have been discussed and included in the conclusions of the present study.

+ We hope that we have addressed all your comments to your satisfaction. We thank you for your time and the improvements your views have made to the manuscript. We remain at your disposal for any further suggestions you may have.

Reviewer 2 Report

Comments to the author

Basic reporting

The study reported the differences in body composition, physical performance, and adherence to the Mediterranean diet between active and sedentary adolescents; and analyzed the differences between active and sedentary adolescents according to the "fat but fit" paradigm. In general, this study aims at an important thematic and one that is of large interest considering the beneficial impact of physical activity. Whilst the study undoubtedly has merit, there are aspects that need clarification, to improve the readability of the manuscript

ABSTRACT

L12-15: Rewrite this sentence. It is too long and difficult to understand.

INTRODUCTION

Basic reporting: Despite a good review of the problem, I consider that it would be appropriate to introduce more information from the studies provided and complement it with the new ones.

1) The introduction section should be further developed on the following topics: the reduction of physical practice and the decrease of physical fitness among adolescents level. This is a global problem and well-analysed topic for the scientific community. I consider that it would be appropriate to enrich this section including the following references:

-          Arboix-Alió, J., Buscà, B., Sebastiani, E. M., Aguilera-Castells, J., Marcaida, S., Garcia Eroles, L., & Sánchez López, M. J. (2020). Temporal trend of cardiorespiratory endurance in urban Catalan high school students over a 20 year period. PeerJ, 8, e10365. https://doi.org/10.7717/peerj.10365

-          Ortega, F. B., Artero, E. G., Ruiz, J. R., Espana-Romero, V., Jimenez-Pavon, D., Vicente-Rodriguez, G., Moreno, L. A., Manios, Y., Beghin, L., Ottevaere, C., Ciarapica, D., Sarri, K., Dietrich, S., Blair, S. N., Kersting, M., Molnar, D., Gonzalez-Gross, M., Gutierrez, A., Sjostrom, M., & Castillo, M. J. (2011). Physical fitness levels among European adolescents: the HELENA study. British Journal of Sports Medicine, 45(1), 20–29. https://doi.org/10.1136/bjsm.2009.062679

-          Ferrari, G. L. D. M., Matsudo, V. K. R., & Fisberg, M. (2015). Changes in physical fitness and nutritional status of schoolchildren in a period of 30 years (1980-2010). Revista Paulista de Pediatria, 33(4), 415–422. https://doi.org/10.1016/j.rppede.2015.03.001

-          Tomkinson, G. R., Lang, J. J., & Tremblay, M. S. (2017). Temporal trends in the cardiorespiratory fitness of children and adolescents representing 19 high-income and upper middle-income countries between 1981 and 2014. British Journal of Sports Medicine, bjsports-2017-097982. https://doi.org/10.1136/bjsports-2017-097982

2) What is the novelty of this study? Research in most studies has shown the importance of physical activity during adolescence. Thus the authors have to find some points that their study provides something new to the literature (and highlight them) and also search deeper in the literature.

MATERIAL AND METHODS

Participants

·       I suggest you improve the description of the sample selection. How have you chosen these samples of adolescents?

·       Which was the physical activity level and/or the sports participation level of those adolescents? Do you have any data to provide? Also, did you calculate the biological maturity of these adolescents? It may have influenced the results of the present study.

·       Please, could you deeply present the demographic data of the sample? Where are the adolescents from? Which region of Spain? From a rural or urban area?

·       A limitation of the present study is the biological maturity of the sample. Do the authors have measured the peak height velocity of the children? If not then they should mention this limitation in the discussion section. 

Measurements

General comment: The main aspect is related to data acquisition. There is needed to present and detail some of the procedures.

Questionnaire measures

·       L 89-98: Why the authors have not chosen the IPAQ adolescents adapted version?

Physical fitness test

·       Please present information about how and where tests were performed (local/school, single day, kind of warm-up performed, who organized and applied the tests, etc.). Please detail all the main information and the necessary details in order to provide the reader a clear picture of how physical tests were performed.

·       Why tests were not randomized? How the fatigue could have affected the results?

·       L 132-138: Is really the Sit and Reach a valid test to measure hamstring and lower back flexibility? Consider this reference: https://pubmed.ncbi.nlm.nih.gov/24570599/

RESULTS

·       The authors should also provide reliability analyses of the tests because they measured adolescents who do not have much experience in fitness tests. Please provide the intraclass correlation coefficients (ICC) and CV in your measured variables.

·       Please provide the most important comparisons of the effect size values.

·       Table 2: Provide the measurement units for the fitness test.

DISCUSSION

General comment: In the discussion, the authors should be more specific to answer the study's aim and address the main results. The authors should search deeper in the literature in order to better explain their results.  Moreover, some grammatical structures are too repetitive (i.e: “These results are in agreement with those found in previous research” difficulting the comprehension.

-          L 293-300: This paragraph should be better discussed. Why did the adolescents' height influence their physical activity? Please develop this assumption and provide some evidence that could explain it. I understand the fact that in some sports, early maturation is a competitive advantage. However, in other sports, the late maturation athletes are who have a better performance.

-          L 305-313: This paragraph should be better discussed.

-          L 376-384: I find really important “the promotion of out-of-school physical activity”, which the authors point out as a strategy to compensate for the reduced hours of PE subject in the Spanish curriculum. However, I think that investigations like this one should be evidence to claim more hours of PE in the Spanish curriculum and also to increase the PE status, which has traditionally been perceived by the educational community as less valuable than other more-traditional academic subjects. I suggest the authors consider this point as a practical implication.

-        I suggest you include more considerations about the limitation of your study.

Author Response

Basic reporting

The study reported the differences in body composition, physical performance, and adherence to the Mediterranean diet between active and sedentary adolescents; and analyzed the differences between active and sedentary adolescents according to the "fat but fit" paradigm. In general, this study aims at an important thematic and one that is of large interest considering the beneficial impact of physical activity. Whilst the study undoubtedly has merit, there are aspects that need clarification, to improve the readability of the manuscript.

 + Dear reviewer, we appreciate your review and comments. We will try to respond to all of them to improve the quality of the manuscript.

ABSTRACT

L12-15: Rewrite this sentence. It is too long and difficult to understand.

+ Thank you for your input. This part of the abstract has been modified.

INTRODUCTION

Basic reporting: Despite a good review of the problem, I consider that it would be appropriate to introduce more information from the studies provided and complement it with the new ones.

1) The introduction section should be further developed on the following topics: the reduction of physical practice and the decrease of physical fitness among adolescents level. This is a global problem and well-analysed topic for the scientific community. I consider that it would be appropriate to enrich this section including the following references:

- Arboix-Alió, J., Buscà, B., Sebastiani, E. M., Aguilera-Castells, J., Marcaida, S., Garcia Eroles, L., & Sánchez López, M. J. (2020). Temporal trend of cardiorespiratory endurance in urban Catalan high school students over a 20 year period. PeerJ, 8, e10365. https://doi.org/10.7717/peerj.10365

- Ortega, F. B., Artero, E. G., Ruiz, J. R., Espana-Romero, V., Jimenez-Pavon, D., Vicente-Rodriguez, G., Moreno, L. A., Manios, Y., Beghin, L., Ottevaere, C., Ciarapica, D., Sarri, K., Dietrich, S., Blair, S. N., Kersting, M., Molnar, D., Gonzalez-Gross, M., Gutierrez, A., Sjostrom, M., & Castillo, M. J. (2011). Physical fitness levels among European adolescents: the HELENA study. British Journal of Sports Medicine, 45(1), 20–29. https://doi.org/10.1136/bjsm.2009.062679

- Ferrari, G. L. D. M., Matsudo, V. K. R., & Fisberg, M. (2015). Changes in physical fitness and nutritional status of schoolchildren in a period of 30 years (1980-2010). Revista Paulista de Pediatria, 33(4), 415–422. https://doi.org/10.1016/j.rppede.2015.03.001

- Tomkinson, G. R., Lang, J. J., & Tremblay, M. S. (2017). Temporal trends in the cardiorespiratory fitness of children and adolescents representing 19 high-income and upper middle-income countries between 1981 and 2014. British Journal of Sports Medicine, bjsports-2017-097982. https://doi.org/10.1136/bjsports-2017-097982

+ Thank you very much for your contributions. The recommended references have been included to give more consistency to the introduction. 

2) What is the novelty of this study? Research in most studies has shown the importance of physical activity during adolescence. Thus, the authors have to find some points that their study provides something new to the literature (and highlight them) and also search deeper in the literature.

+ Thank you very much for your great contribution. A final paragraph has been included with the previous limitations and the reasons for conducting this study.

MATERIAL AND METHODS

Participants

I suggest you improve the description of the sample selection. How have you chosen these samples of adolescents?

+ Thank you for your input. A more detailed description and flowchart of the selection has been included.

Which was the physical activity level and/or the sports participation level of those adolescents? Do you have any data to provide? Also, did you calculate the biological maturity of these adolescents? It may have influenced the results of the present study.

+ Thank you for your comment. Information on the average level of physical activity performed has been included in the "participants" section. Biological maturation has also been introduced as a covariate of the model and discussed in the "discussion" section.

Please, could you deeply present the demographic data of the sample? Where are the adolescents from? Which region of Spain? From a rural or urban area?

+ Thank you very much. Done.

A limitation of the present study is the biological maturity of the sample. Do the authors have measured the peak height velocity of the children? If not then they should mention this limitation in the discussion section.

+ Thank you very much for your contribution. The maturity offset of the children has been calculated and biological maturation has been introduced as a covariate of the model. The formulas for the calculation have been specified in the method and the results obtained have been discussed.

Measurements

General comment: The main aspect is related to data acquisition. There is needed to present and detail some of the procedures.

+ Thank you for your comment. We hope we have responded to the issues you have raised.

Questionnaire measures

L 89-98: Why the authors have not chosen the IPAQ adolescents adapted version?

+ Thank you for your comment. We know that the large number of questionnaires makes the choice of a questionnaire a sometimes-conflicting decision. Our decision was based on the fact that in the case of adolescents aged 14 years or younger, the correlations of the IPAQ questionnaire for adolescents are unsatisfactorily low (Hagstromer et al., 2008) and, considering that our sample presents an average age of 14 years, we thought it more convenient to use the PAQ-A, which had been previously validated in a sample of Spanish adolescents, showing adequate validity and reliability for measuring physical activity (Martínez-Gómez et al., 2009).

Physical fitness test

Please present information about how and where tests were performed (local/school, single day, kind of warm-up performed, who organized and applied the tests, etc.). Please detail all the main information and the necessary details in order to provide the reader a clear picture of how physical tests were performed.

+ Thank you very much for your contribution. All information regarding has been included in "procedure" section.

Why tests were not randomized? How the fatigue could have affected the results?

+ According to the recommendations of the National Strength and Conditioning Association (NSCA) and previous research, the sit-and-reach test should be performed before the warm-up due to the improvement it produces in performance, therefore, it should be the first test. The 20-m shuttle run test requires the maximum exhaustion of the participants, so it needs more recovery time and, therefore, it is performed last so that it does not affect the performance of the other tests. The physical and metabolic demands of the remaining tests (handgrip, sprint and CMJ) were minimal, as was the fatigue generated, thanks to the rest time provided between each repetition and each test. These three tests were performed randomly in adolescents. All these information and reasons has been included in the manuscript in “procedure” section.

L 132-138: Is really the Sit and Reach a valid test to measure hamstring and lower back flexibility? Consider this reference: https://pubmed.ncbi.nlm.nih.gov/24570599/

+ Thank you for your comment. According to the reference you indicate (Mayorga-Vega et al., 2014) and the study with adolescent population by Castro-Piñero et al. (2009), we have modified the information in the text, leaving only that this test is valid for measuring hamstring flexibility. In addition, the citation has been included in the manuscript. Thank you very much for this important contribution.

RESULTS

The authors should also provide reliability analyses of the tests because they measured adolescents who do not have much experience in fitness tests. Please provide the intraclass correlation coefficients (ICC) and CV in your measured variables.

+ Thank you very much for your comment. Both statistics have been calculated for the physical fitness variables that were measured twice and has been included in results section.

Please provide the most important comparisons of the effect size values.

+ Thank you for your input. Table 4 has been included in which the pairwise comparisons are presented, including the p-value, effect size, 95% CI and mean differences for each.

Table 2: Provide the measurement units for the fitness test.

+ Thank you very much for your contribution. The units of measurement of the physical tests have been included in all tables.

DISCUSSION

General comment: In the discussion, the authors should be more specific to answer the study's aim and address the main results. The authors should search deeper in the literature in order to better explain their results. Moreover, some grammatical structures are too repetitive (i.e: “These results are in agreement with those found in previous research” difficulting the comprehension.

+ Thank you for your comment. The discussion has been revised, including more citations, addressing the results in more depth and avoiding repetitive grammatical structures. 

L 293-300: This paragraph should be better discussed. Why did the adolescents' height influence their physical activity? Please develop this assumption and provide some evidence that could explain it. I understand the fact that in some sports, early maturation is a competitive advantage. However, in other sports, the late maturation athletes are who have a better performance.

+ Thank you for your comment. The whole paragraph has been rewritten providing the evidence to justify the results found.

L 305-313: This paragraph should be better discussed.

+ Thank you for your comment. The discussion of this paragraph has been expanded.

L 376-384: I find really important “the promotion of out-of-school physical activity”, which the authors point out as a strategy to compensate for the reduced hours of PE subject in the Spanish curriculum. However, I think that investigations like this one should be evidence to claim more hours of PE in the Spanish curriculum and also to increase the PE status, which has traditionally been perceived by the educational community as less valuable than other more-traditional academic subjects. I suggest the authors consider this point as a practical implication.

+ Thank you for your excellent input. We fully agree with you and have included this in the practical implications of this study.

I suggest you include more considerations about the limitation of your study.

+ Thank you for your comments. The limitations of this study have been extended.

+ We hope that we have addressed all your comments to your satisfaction. We thank you for your time and the improvements your views have made to the manuscript. We remain at your disposal for any further suggestions you may have.

Round 2

Reviewer 1 Report

Most of the comments were properly accepted in the revised version of the manuscript. I would like to suggest one minor comment to add the percentage value from the odds ratio. Because the authors can interpret how much greater or lower the specific values between active and sedentary groups. It could be easier to describe the severity of the risks in the discussion section. 

Author Response

#Reviewer 1

Most of the comments were properly accepted in the revised version of the manuscript. I would like to suggest one minor comment to add the percentage value from the odds ratio. Because the authors can interpret how much greater or lower the specific values between active and sedentary groups. It could be easier to describe the severity of the risks in the discussion section. 

+ Thank you very much for your comment. Since they are not categorical variables, we have not been able to calculate percentages. If we are wrong and it can be done, please tell us how, as we do not know, and we will include it immediately.

Reviewer 2 Report

General comment:

The authors have significantly improved the quality of the manuscript. However, there is a main concern regarding data.

In particular, you reported a high %CV for the fitness test variables, which may make it quite difficult to be confident. Based on the threshold <10% for the CV values and the recommendations of Koo and Li, some of these fitness tests are not reliable and therefore should be excluded from further analysis. Based on the fact that some of the variables do not meet the thresholds for reliability and variability, many of the comparisons should not have been made.

You therefore may be recommended to determine why these data is so variable and find ways to improve the between trial variability or determine if this method is even appropriate.

Other comments:

The tables are containing too much data information, as well as poor readability. I recommend expressing some of the table information utilizing figures.

Please, double-check and correct the duplicated references. Correct it also throughout the text.

Author Response

#Reviewer 2

General comment:

The authors have significantly improved the quality of the manuscript. However, there is a main concern regarding data.

+ Thank you very much.

In particular, you reported a high %CV for the fitness test variables, which may make it quite difficult to be confident. Based on the threshold <10% for the CV values and the recommendations of Koo and Li, some of these fitness tests are not reliable and therefore should be excluded from further analysis. Based on the fact that some of the variables do not meet the thresholds for reliability and variability, many of the comparisons should not have been made. You therefore may be recommended to determine why these data is so variable and find ways to improve the between trial variability or determine if this method is even appropriate.

+ Dear reviewer, thank you very much for your comment. We fully understand what you indicate and the importance of CV values <10%. For this reason, we have repeated this statistical analysis again, as the values seemed excessively high to us. In doing so, we have realized that we made an error in the previous version of the manuscript when converting the value to percentage. This value has been modified in the text.

If you wish, we can attach the database to check the veracity of these analyses.

Other comments:

The tables are containing too much data information, as well as poor readability. I recommend expressing some of the table information utilizing figures.

+ Thank you for your input. We have converted Table 3 into a Figure, as it is clearer.

Please, double-check and correct the duplicated references. Correct it also throughout the text.

+ Thank you for your input. The references have been corrected, eliminating duplicate references.